# Tool Wear Condition Monitoring by Combining Variational Mode Decomposition and Ensemble Learning

**DOI:** 10.3390/s20216113

**Published:** 2020-10-27

**Authors:** Jun Yuan, Libing Liu, Zeqing Yang, Yanrui Zhang

**Affiliations:** 1School of Mechanical Engineering, Hebei University of Technology, Tianjin 300130, China; 201711201028@stu.hebut.edu.cn (J.Y.); yangzeqing@hebut.edu.cn (Z.Y.); 2Experimental Training, Hebei University of Technology, Tianjin 300401, China; 2008021@hebut.edu.cn

**Keywords:** tool wear condition monitoring, spindle motor current, time–domain analysis, frequency–domain analysis, variational mode decomposition (VMD), ensemble learning (EL)

## Abstract

Most online tool condition monitoring (TCM) methods easily cause machining interference. To solve this problem, we propose a method based on the analysis of the spindle motor current signal of a machine tool. Firstly, cutting experiments under multi-conditions were carried out at a Fanuc vertical machining center, using the Fanuc Servo Guide software to obtain the spindle motor current data of the built-in current sensor of the machine tool, which can not only apply to the actual processing conditions but, also, save costs. Secondly, we propose the variational mode decomposition (VMD) algorithm for feature extraction, which can describe the tool conditions under different cutting conditions due to its excellent performance in processing the nonstationary current signal. In contrast with the popular wavelet packet decomposition (WPD) method, the VMD method was verified as a more effective signal-processing technique according to the experimental results. Thirdly, the most indicative features that relate to the tool condition were fed into the ensemble learning (EL) classifier to establish a nonlinear mapping relationship between the features and the tool wear level. Compared with existing TCM methods based on current sensor signals, the operation process and experimental results show that using the proposed method for the monitoring signal acquisition is suitable for the actual processing conditions, and the established tool wear prediction model has better performance in both accuracy and robustness due to its good generalization capability.

## 1. Introduction

Compared with the traditional manufacturing industry, the advanced manufacturing industry is moving towards intelligent development; however, there are many problems to be solved in the development of modern manufacturing. One problem is whether the tool status can be accurately monitored in real time. Tool wear is a common problem in precision manufacturing, which affects the productivity of modern high-speed computer numerical control (CNC) manufacturing greatly [1]. Severe tool wear may lead to not only scrapped components but, also, possible damage to the machine tool; therefore, it is necessary to establish a reliable and effective tool condition monitoring (TCM) system.

The main TCM methods are direct and indirect, respectively. Since the direct methods need to be shut down to measure the tool wear value, they cannot meet the needs of industrial applications. The indirect methods analyze relevant information collected from one or more sensors to estimate tool wear conditions by using a machine-learning technique. Since the indirect methods can implement online monitoring, it has been widely adopted. The most commonly used indirect monitoring signals include cutting force signal [2,3,4], vibration signal [5,6,7], acoustic emission signal [8,9], machined surface image [10,11], and current signal [12,13,14,15,16,17].

Although the change of cutting force is closely related to tool wear conditions, the installation of the force measurement system on machine tools is difficult or unfeasible during their machining. In addition, the force measurement system is expensive—these shortcomings diminish its usefulness in real industrial applications. Compared to the force measurement system, the vibration sensor can be installed easily on machine tools. Due to the source of the vibration signal, including the impact of cutting in and out, the friction between the workpiece and cutting tool, the chip fracture and the vibration of the machine tool, and so on, vibration information that is not related to the tool wear condition is difficult to filter. In addition, the position of sensor installation affects the monitoring result greatly. Due to this, it is difficult to use the vibration signal to monitor the tool wear state accurately. Compared with the vibration signal, the frequency of the acoustic emission signal is higher, and the interference of low-frequency vibration can be eliminated; however, the acoustic emission signal may exhibit nonstationary characteristics, even if the tool state and cutting parameters are unchanged. The chip fracture and the tool’s cutting in and out can affect the acoustic emission signal; therefore, the preprocessing and feature extraction of the acoustic emission signal related to the tool wear condition is exceedingly difficult. Since the characteristics of the machined surface image are closely related to the tool wear state, the feature extraction of the machined surface image can be used to judge the tool wear status. The machined surface image can be obtained by a vision-measuring instrument. For example, an industrial camera can be applied to capture the machined surface image. However, considering the influence of the cutting fluid and chips, it is difficult to achieve accurate online TCM.

Compared with the above sensors, the motor current sensor is considered to be more suitable for the actual processing environment due to its relatively simple application and lack of installation effect on machining operations [14]. In addition, the cost of the current sensor is cheap, and it is easy to obtain the current signal related to the tool wear state. Further, the internal system of different types of machine tools is equipped with an embedded Ethernet board card, so that users can easily connect to a computer via an Ethernet cable to obtain the current data. Especially for some open numerical control machine tools, the user can easily read the current data of the built-in current sensor through the secondary development protocol; therefore, tool wear condition monitoring based on the current signal analysis can both save costs and apply to processing environments; however, the single current sensor is less commonly used than above other types of sensors.

Although multiple sensors can get more information related to the tool wear condition than a single current sensor, the costs and the interference caused by sensors increases with an increasing amount of sensors; therefore, in recent years, some researchers have made some achievements in the application of single-current sensors for tool wear monitoring. For example, Akbari et al. [12] indicated that the total harmonic distortion (THD) and crest factor measures of the current signal can detect tool wear degree; however, they need additional harmonic analysis equipment to calculate the THD. Further, the tool wear condition will be misjudged if the tool cuts in or cuts out of the workpiece or if the power supply system is affected by the interference signal. Mahmoud et al. [13] utilized generalized features of the current signal to describe the tool condition, although the method proposed by them is only used to realize the binary classification of the tool wear condition, even though the feature ranking and selection is also needed. Zhou et al. [14] proposed a TCM method based on the time, frequency, and time–frequency domains of the current signals and an identification model based on an improved kernel extreme learning machine (KELM); however, it is difficult to determine the optimal value of the two output weight vectors when the improved KELM is used for classification. Khajavi et al. [15] used a multi-layer neural network to predict tool wear based on an analysis of the motor current signal. Since the proposed method needs an extremely rich sample for model training, it will waste a lot of manpower and material resources. Lin et al. [16] proposed an intelligent tool breakage monitoring methodology with spindle motor current signals in the milling processing; however, they cannot achieve a multi-classification of tool wear conditions. Based on the current sensor signals, Huang et al. [17] proposed a method of discriminant diffusion maps analysis for evaluating the tool wear status during the milling process; however, the method they proposed needs calculations of the feature dimension reduction and feature fusion. Compared to the above-mentioned TCM methods, this paper proposes a more effective and convenient TCM method, which can avoid the problems of the above monitoring methods. 

As one of the most commonly used signal-processing techniques, the wavelet packet decomposition (WPD) conducts a multi-level band division over the entire signal band and further decomposes the high-frequency band to increase the frequency resolution [14]. Since the selection of the wavelet basis function has a great influence on the decomposition result of the signal, the optimal selection of the wavelet basis function and its parameters is difficult; however, variational mode decomposition (VMD) has obvious advantages in analyzing the nonstationary signals due to its unique decomposition principle. 

For example, VMD has been used to extract features from the nonstationary Bluetooth transient signals to improve classification accuracy. According to the classification results, a higher classification performance is achieved when higher-order statistical features are extracted from band-limited modes in the implementation of VMD in the radio frequency fingerprinting of Bluetooth devices [18]. Ruicheng et al. [19] used the VMD algorithm to extract the low-frequency displacement of the global navigation satellite system. According to the simulation and measured data, the VMD algorithm can effectively resist the modal aliasing caused by noise and discontinuous signals compared to the commonly used empirical mode decomposition (EMD) algorithm. Sahani et al. [20] applied the VMD to the real-time identification of power quality events in electrical power systems. Through the decomposition of the harmonic signal and flicker signal in power systems, the features for power quality identification were obtained. The experimental results showed that this method has a strong, robust antinoise performance and lesser computational complexity. Xiao et al. [21] used the VMD algorithm to decompose the original surface electromyogram signal into multiple variational mode functions (VMFs) and calculate the corresponding composite permutation entropy index (CPEI) of each signal component, and these CPEI features are applied to recognize hand actions. The experimental results show that the method based on VMD and CPEI for hand motion classification is feasible and accurate. Liu et al. [22] adopted the VMD algorithm to extract the time–frequency domain characteristic features of the arc current signals. Compared with some traditional modal decomposition algorithms, the decomposition results show that VMD can obtain high-quality frequency bands by avoiding the modal aliasing; therefore, in this study, we implement the VMD to decompose a nonstationary current signal. 

The commonly used classifiers for tool wear identification include artificial neural networks (ANNs) [5,23], support vector machines (SVM) [24], hidden Markov models (HMM) [25,26], etc. The widely used ANNs are three-layer ANNs. Three-layer ANNs can approach any nonlinear function with arbitrary precision when the number of hidden layers is enough. The disadvantage of ANNs is that, when the sample is insufficient or the sample noise is large, it is easy to appear underfitting or overfitting. The theory of the SVM algorithm is based on the statistical learning theory; the proposed algorithm embodies the structural risk minimization principle, and it has a distinct advantage in analyzing and classifying small-scale samples. When there are only limited labeled samples, SVM has a better generalization than ANNs; however, the penalty actor and the kernel function, along with its parameters, are the main factors affecting the performance of the SVM classification performance. If the parameter selections are improper, the generalization ability of the classifier will be weakened to a large extent. An HMM is a statistical model that assumes the observation sequence is generated by a Markov process with hidden states; therefore, an HMM is a typical generative model. Compared with ANNs, SVM, and other discriminative learning models, the advantage of the generative model is that it has a stronger generalization capability; however, to obtain accurate recognition results, the generative model needs a large number of samples to study. Since the recognition effect of a single classifier is often not ideal, the advantages of ensemble learning (EL) are reflected. EL is a methodology that combines multiple individual learners and can obtain better classification accuracy and generalization ability. The random forest (RF) as a classifier developed by Breiman [27] is an EL classifier.

Many application cases have proved that the RF algorithm has higher prediction accuracy. Wu et al. [6] applied RF to tool wear prediction, as well as compared the performance of RF with ANNs and SVM. The experimental results showed that RF can generate more accurate recognition accuracy than ANNs and SVM. LI et al. [28] proposed a feasible driver identification method utilizing a machine-learning algorithm with driving information. Four basic classification algorithms were performed on the datasets for comparison. The experimental results showed that the RF algorithm had the best performance on identification accuracy among the four basic algorithms. William et al. [29] produced a dataset that contains patterns of radio wave signals obtained using software-defined radios to establish if a subject is standing up or sitting down as a test case. They used the RF model to conduct a real-time classification of a standing or sitting state based on the dataset. The results showed that the RF model can generate 96.70% classification accuracy. 

The remainder of this paper is organized as follows: Section 2 presents the method of spindle motor current signal acquisition, the method of current signal feature extraction, and the tool wear condition prediction based on ensemble learning (EL). Section 3 presents the experimental results and discussion. Section 4 concludes this paper with a summary of the contributions made.

## 2. The Proposed Method

### 2.1. The Method of Spindle Motor Current Signal Acquisition

Since the cutting experiment environment was a Fanuc vertical machining center, and Fanuc Servo Guide is a system debugging software, which can obtain the current amplitude of the spindle motor by setting the relevant conversion coefficient and conversion reference, the Fanuc Servo Guide software can be used to obtain the spindle motor current. 

One of the collected current signals based on Fanuc Servo Guide software is shown in Figure 1. 

Firstly, the Fanuc Servo Guide software was installed on the upper computer (the computer used to receive the spindle current signal). Then, the upper computer communicates with the CNC machine tool through the ethernet-based transmission control protocol and internet protocol. As shown in Figure 2, the Fanuc Pcmcia Lan Card is an external network card; one end of it is connected to the Fanuc vertical machining center, and the other end is connected to the upper computer through an Ethernet cable.

The model of the Fanuc vertical machining center is CY-VMC850 (Yunnan CY Group Co., Ltd., Kunming, China), and the system is the Fanuc 0i-Mate MC system (Beijing Fanuc, Beijing, China). The upper computer can read the internal data of the machine tool through the Ethernet interface of the system. Finally, after setting the sampling time and sampling frequency, the spindle motor current can be obtained by setting the relevant parameters of the current signal acquisition channel on the Fanuc Servo Guide debugging software. 

Compared with using the external current sensor to obtain the current, the upper computer directly reads the current data of the built-in current sensor of the machine tool through the transmission control protocol and internet protocol based on the Ethernet, which does not need to change the structure of the machine tool and does not affect the dynamic machining performance of the spindle. As shown in Figure 2, only one Ethernet cable is needed to connect the upper computer and CNC machine tool, which avoids the problem of a wiring complex affecting the processing; therefore, the method of obtaining the current in this paper is convenient and cost-effective. 

### 2.2. The Method of Current Signal Feature Extraction

#### 2.2.1. Feature Extraction Based on the Time–Domain Analysis

The signal with time as its independent variable is the most basic and intuitive form of expression. As the most basic detection method, the time–domain analysis can detect the status of some devices quickly and effectively.

There is a correlation between the degree of tool wear and time–domain statistics features of the current signal. For example: under the same working conditions, the root mean square value of the alternating current (AC) signal increases with the increase of the tool wear degree; therefore, the statistical characteristics of the spindle motor current related to tool wear can be extracted by the time–domain analysis, and these statistical features can be analyzed and calculated to identify the tool wear condition to a certain extent.

Figure 3 shows the time domain waveform of the current signal in different tool wear conditions: 

The output power of the spindle motor changes with the variation of tool wear degree, which is reflected in the spindle motor current signal. Figure 3 shows that the time–domain waveform of the current signal in different tool wear conditions are also different. 

In our study, the time–domain statistics features of the current signal sensitive to tool wear were extracted as follows: mean value, root mean square value, kurtosis factor, and margin factor.

#### 2.2.2. Feature Extraction Based on Frequency–Domain Analysis 

Unlike the time–domain analysis, the frequency–domain analysis is performed to show the energy distribution of different frequency bands in the frequency–domain. There is also a correlation between the tool wear degree and frequency–domain statistics features of the current signal. For example, the frequency centroid reflects the position change of the spectrum centroid, the mean square frequency reflects the position change of the main frequency band of the signal, and the frequency variance reflects the energy distribution of a different spectrum. These frequency–domain statistics features change with the variation of tool wear degree. 

Figure 4 shows the amplitude–frequency chart of the current signal in different tool wear conditions: 

The frequency of the cutting vibration varies with the change of tool wear loss, which will be reflected in the frequency–domain of the current signal. Figure 4 shows that the frequency–amplitude characteristics in different tool wear conditions are also different. 

In this study, the frequency–domain statistics features of the current signal sensitive to tool wear are extracted as follows: frequency centroid, mean square frequency, and frequency variance. 

#### 2.2.3. Feature Extraction Based on Variational Mode Decomposition (VMD)

The signal can be decomposed into several modal components by VMD. In the process of obtaining modal components, the frequency center and bandwidth of each modal component are determined by the iterative search for the optimal solution of the variational model, so that the signal can be adaptively divided in the frequency–domain and each component can be separated effectively. 

The constraint decomposition model of VMD can be expressed as
(1)$$\underset{\left\{u_{k}\right\},\left\{\omega_{k}\right\}}{\min}\left\{{\sum_{k=1}^{K}\left\|\partial_{t}\left[\left(\delta \left(t\right)+\frac{j}{\pi t}\right)*u_{k}\left(t\right)\right]e^{-j\omega_{k}t}\right\|}_{2}^{2}\right\}\;s_{\cdot }t_{\cdot }\sum_{k=1}^{K}u_{k}=f$$
where f is the input signal, $\left\{u_{k}\right\}=\left\{u_{1},\cdots ,u_{K}\right\}$ is the different modal components from decomposition, and $\left\{\omega_{k}\right\}=\left\{\omega_{1},\cdots ,\omega_{K}\right\}$ are the frequency center of each modal component. To find the optimal solution of the constrained variational in Equation (1), the quadratic penalty factor and the augmented Lagrangian function are introduced, respectively, namely
(2)$$\begin{array}{l} L\left(\left\{u_{k}\right\},\left\{\omega_{k}\right\},\lambda \right)=\alpha {\sum_{k=1}^{K}\left\|\partial_{t}\left[\left(\delta \left(t\right)+\frac{j}{\pi t}\right)*u_{k}\left(t\right)\right]e^{-j\omega_{k}t}\right\|}_{2}^{2}\\ +{\left\|f\left(t\right)-\sum_{k=1}^{K}u_{k}\left(t\right)\right\|}_{2}^{2}+\langle \lambda \left(t\right),f\left(t\right)-\sum_{k=1}^{K}u_{k}\left(t\right)\rangle \end{array}$$
where ∂ is the penalty factor, and λ is the Lagrange multiplication operator. The alternating direction multiplier method is used to solve the problem, and the saddle point of the augmented Lagrangian function can be gained by alternately refreshing $u_{k}^{n+1}$, $\omega_{k}^{n+1}$, and $\lambda^{n+1}$. The saddle point is the optimal solution of the constrained variational model.

The iterative formula $u_{k}^{n+1}$ can be expressed as
(3)$$u_{k}^{n+1}=\underset{u_{k}\in X}{\arg\min}\left\{\begin{array}{l} \alpha {\left\|\partial_{t}\left[\left(\delta \left(t\right)+\frac{j}{\pi t}\right)*u_{k}\left(t\right)\right]e^{-j\omega_{k}t}\right\|}_{2}^{2}\\ +{\left\|f\left(t\right)-\sum_{i}u_{i}\left(t\right)+\frac{\lambda \left(t\right)}{2}\right\|}_{2}^{2} \end{array}\right\}$$

In Equation (3), $\omega_{K}$ is equal to $\omega_{k}^{n+1}$, and $\sum_{i}u_{i}\left(t\right)$ is equal to $\sum_{i\neq k}u_{i}{\left(t\right)}^{n+1}$. Equation (3) is transformed into the frequency–domain for the solution, and the frequency–domain update expression of each modal component is obtained as follows: (4)$${\hat{u}}_{k}{}^{n+1}\left(\omega \right)=\frac{\hat{f}\left(\omega \right)-\sum_{i\neq k}{\hat{u}}_{i}\left(\omega \right)+\frac{\hat{\lambda }\left(\omega \right)}{2}}{1+2\alpha {\left(\omega -\omega_{k}\right)}^{2}}$$

Similarly, the center frequency in the frequency–domain is obtained, and the updated expression of $\omega_{k}$ is obtained as follows:(5)$$\omega_{k}{}^{n+1}=\frac{\int_{0}^{\infty }\omega {\left|{\hat{u}}_{k}\left(\omega \right)\right|}^{2}d\omega }{\int_{0}^{\infty }{\left|{\hat{u}}_{k}\left(\omega \right)\right|}^{2}d\omega }$$
where ${\hat{u}}_{k}{}^{n+1}\left(\omega \right)$ is considered the Wiener filter of the current residual $\hat{f}\left(\omega \right)-\sum_{i\neq k}{\hat{u}}_{i}\left(\omega \right)$, and $\omega_{k}^{n+1}$ is the power spectral centroid of the current mode function. $\left\{u_{k}\left(t\right)\right\}$ can be obtained by the inverse Fourier transformation of $\left\{{\hat{u}}_{k}\left(\omega \right)\right\}$. 

According to the principle of VMD, the algorithm of VMD is to update all modal components in the frequency–domain and then transform them into the time–domain by inverse Fourier transformation. The procedure of the algorithm is as follows: 
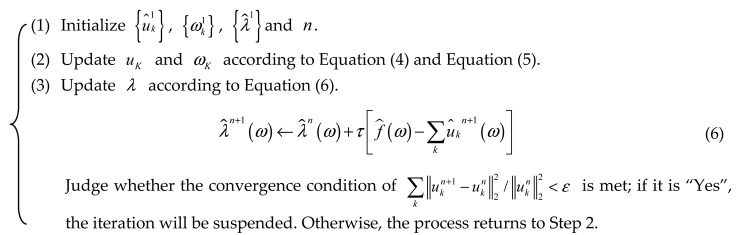


The algorithm avoids the process of getting the local mean curve by using extreme points in the recursive adaptive decomposition algorithm, which results in evident inhibition on the phenomena of mode mixing among intrinsic mode function components after the empirical mode decomposition.

Mode mixing means that similar characteristic time scales are distributed in different modal components. For example, adjacent modal components contain waveforms with the same or similar frequency components. In this study, the number of modal components was determined by combining the center frequency and waveform of the current signal after VMD. The results of the VMD show that the phenomenon of mode mixing occurs when the center frequency of each mode was close to each other or unevenly distributed.

For one current signal sample decomposed by VMD, the central frequencies of the modal components corresponding to the number of different modal components are shown in Table 1. 

For one current signal sample, the current signal waveforms after VMD under different K values are shown in Figure 5.

When using VMD to decompose signals, the number of modal components should be set in advance. If the set number is too small, the signal cannot be effectively decomposed, which will affect the extraction of effective features. If the number is too big, it will cause mode mixing and also affect the extraction of effective features; therefore, it is necessary to ensure that the signal is decomposed sufficiently and that mode mixing cannot occur. Table 1 and Figure 5 both show that the signal cannot be decomposed sufficiently when k is 2, 3, or 4; when k is 6, the phenomenon of mode mixing begins to appear in the red box in Figure 5e; when k is 7, the phenomenon of mode mixing becomes more obvious, as shown in the red box in Figure 5f. As such, we set K to 5 in our experiment. Figure 6 shows the results of the VMD chart of the current signal in different tool wear conditions.

Figure 6 shows that the total energy of the frequency band obtained by the VMD of the current signal is different under different tool wear conditions, and the energy distribution reflected in different frequency bands is also different. In this study, we extracted the energy of different frequency bands of the current signal after VMD was performed to be used as the sensitive features, in order to identify the tool wear conditions. Since the K value was 5, five dimensions of the energy statistical features can be obtained.

#### 2.2.4. Extracted Statistical Features Based on the Proposed Method

The time–domain, frequency–domain, and VMD statistical features listed in Table 2 were extracted from the original spindle motor current signals.

In the time–domain analysis, four types of statistical features were extracted. The mean ($x_{i}$) represents the central tendency of the spindle motor current signal. The root mean square (RMS) shows the average energy of the spindle motor current signal within a given time interval. The Kurtosis factor (Kur) represents the transient phenomena and stationarity of the spindle motor current signal. The margin factor (Mar) represents the ratio of the square root amplitude value to the peak value in the spindle motor current signal. $x_{\max}$ represents the maximum instantaneous current value of the signal. In the frequency–domain analysis, three statistical features that quantify the frequency property of the spindle motor current signal were extracted. $f_{i}$ is the instantaneous value of frequency of the frequency spectrum that converted from the original signal by using a fast Fourier transform, $p\left(f_{i}\right)$ is the instantaneous amplitude of the Fourier spectrum, and N is the number of sample points. VMD was utilized to effectively decompose the original current signal into five modal components. The frequency band energy of each modal component can provide more detailed information, which reflects the tool wear condition from the perspective of the time–frequency domain. $x_{ik}$ represents the amplitude of discrete points in each modal component. Accordingly, five statistical features can be extracted from each current signal by VMD. Thus, a total of 12 statistical features can be obtained by time–domain analysis, frequency–domain analysis, and VMD.

### 2.3. The Pattern Recognition Based on Ensemble Learning (EL)

The random forest (RF) algorithm is an EL method that builds a forest of decision trees from bootstrap samples of a training dataset. Every decision tree produces a response, given a group of predictor values. In every decision tree, each internal node represents a test on an attribute, each branch represents the outcome of the test, and each leaf node represents a class label for classification. Based on each decision tree and voting rules, the final classification result of RF is obtained.

Figure 7 shows a detailed RF algorithm.

Since the training samples are randomly selected when the decision tree is generated, the optimal attributes when the nodes split in the decision tree are also randomly selected; taking into account the stochastic characteristics of the training samples and the randomness of feature selection, RF can ensure that it does not produce overfitting phenomenon in the classification.

Many of the application cases described in Section 1 have shown that the RF algorithm has good tolerance to outliers and noise, can avoid overfitting, and has better generalization ability and higher prediction precision.

Given the excellent recognition ability of RF, we implemented an RF classifier to identify the tool wear condition. The pattern recognition route is shown in Figure 8.

## 3. Experimental Results and Discussions

### 3.1. Experimental Setup

The experimental setup is shown in Figure 9.

The workpiece material and cutter material used in the experiment were low carbon steel and high-speed steel, respectively. The length, width, and height of the workpiece are 200, 100, and 40 mm, respectively. Cutting experiments under multi-conditions were conducted on a three-axis high-speed CNC machine (CY-VMC850). Table 3 shows the operating conditions.

Since the material of the workpiece and cutter used in the experiment were low-carbon steel.

The spindle motor current data were monitored in real time during each cutting test. The sampling frequency was 1000 Hz.

During the test, new cutting tools were used to carry out the cutting test under experimental conditions. After each cutting test, the value of the tool wear was measured off-line using a microscope. Figure 10 shows the tool images of different wear stages.

Since the material of the workpiece and cutter used in the experiment were low-carbon steel and high-speed steel, respectively, and the processing condition was rough machining, the tool flank’s maximum width of the wear land (VB) reaching or above 0.5 mm was considered worn. The range of the tool wear value in different wear stages is shown in Table 4.

After the tool wear measurements, three different tool wear stages were obtained.

A complete tool wear monitoring system includes the acquisition of monitoring data, feature extraction, and pattern recognition. Above all, the spindle motor current data collected by Fanuc Servo Guide software formed the raw data. The raw data were transformed into a set of statistical features supported by a machine-learning algorithm. The statistical features were input into the RF algorithm to identify the tool wear conditions.

### 3.2. Results and Discussions

In machine learning, two-thirds of the input data samples were selected at random for model training. The remaining input data samples were used for model testing.

In the process of tool wear recognition using RF, since the selection of training samples was random, it was necessary to carry out multiple classifications to evaluate the performance of the identification model. When the RF was used for classification, classification was repeated 10 times, the average value of 10 classification results was taken as the final classification result, and the variance of the 10 times identification results was calculated to evaluate the performance of the identification model.

Under nine different experimental conditions, nine sets of tool wear condition recognition accuracy results were obtained. As an example, Table 5 and Table 6 show the predicted recognition accuracy of tool wear conditions with different feature extraction methods using wavelet packet decomposition (WPD) and VMD, respectively.

Table 5 shows the recognition accuracy of the tool wear condition based on the time–domain analysis, frequency–domain analysis, and WPD.

Table 6 shows the recognition accuracy of the tool wear condition based on the time–domain analysis, frequency–domain analysis, and VMD.

Table 5 and Table 6 show that the recognition accuracy of the tool wear condition based on the time–domain analysis, frequency–domain analysis, and VMD feature extraction is 7.74% higher than the recognition accuracy of the tool wear condition based on the time–domain analysis, frequency–domain analysis, and WPD feature extraction. Further, the recognition model based on VMD has less variation in the recognition results—that is to say, its recognition model has better robustness.

Table 7 and Table 8 show the predicted recognition accuracy of the tool wear condition under different operating conditions using WPD and VMD, respectively.

Table 7 and Table 8 show that the statistical features of the current signal obtained by VMD are more sensitive to tool wear than those obtained by WPD—that is to say, the performance of VMD is better than WPD in extracting statistical features of the current signal sensitive to tool wear.

We know that various classifiers can obtain various characteristics in the area of pattern recognition. In this paper, the radial basis function (RBF)ANNs, SVM, and RF were used to identify the tool wear conditions, respectively. In this paper, the performance of the three classifiers was evaluated by the variance of the identification results and the recognition accuracy.

Figure 11 and Figure 12 show the predicted recognition accuracy and variance under different conditions with different classifiers using RBFANNs, SVM, and RF, respectively.

Figure 11 and Figure 12 show that the recognition accuracy of the tool wear condition based on RF is higher than that based on RBFANNs or SVM. In addition, it has a smaller variance in its recognition result. The above two points confirm that the EL classifier has an excellent performance in tool wear condition prediction.

## 4. Conclusions

In the existing research papers of online tool wear condition recognition, most researchers obtain the monitoring information related to tool wear through external sensors such as a stationary dynamometer, acoustic emission sensor, piezo accelerometers, industrial camera, and so on. Although they can realize the identification of tool wear conditions, the installation of these external sensors causes machining interference. Given this serious shortcoming, we propose a TCM method that can be applied to actual processing conditions and has a high prediction accuracy. Our work is summarized as follows:

(1) We provided a deep review of past TCM methods. The analysis and comparison show that TCM based on the spindle motor current signal is an affordable, feasible, and effective monitoring method. Then, through experimentation, we compared our proposed method and other TCM methods based on the current signal analysis. The experimental results show that using our proposed approach for TCM is more suitable and effective for the actual processing environment.

(2) The time–domain analysis, frequency–domain analysis, and VMD were jointly used to extract the sensitive features of the monitoring current signal, which is closely related to the tool wear condition. The experimental results show that the VMD has an excellent performance in processing the nonstationary current signal; the frequency band and energy features obtained by the VMD of the current signal are more sensitive to tool wear than that obtained by the frequently used WPD of the current signal.

(3) Since RF can handle thousands of input variables without variable deletion, it improves the identification efficiency. Due to this, we chose RF for tool wear condition prediction. The prediction model was established by combining the sensitive features and RF. The results show that the established tool wear prediction model has a higher prediction accuracy, smaller variance, and better robustness.

## Figures and Tables

**Figure 1 sensors-20-06113-f001:**
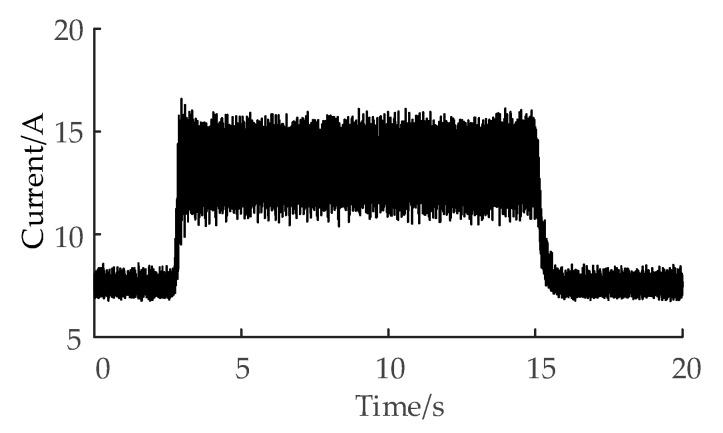
Collected current signal based on the Fanuc Servo Guide software.

**Figure 2 sensors-20-06113-f002:**
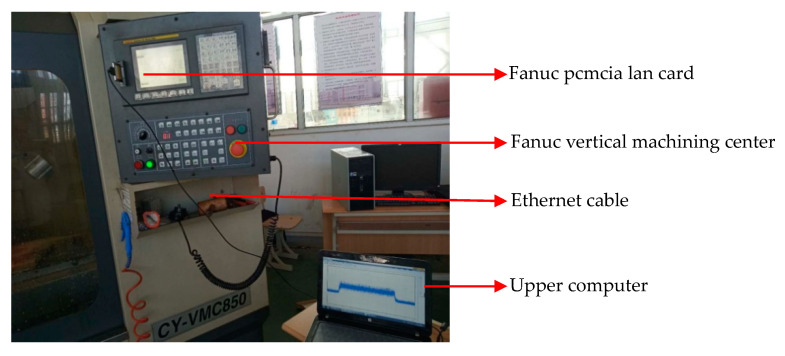
Current signal acquisition diagram.

**Figure 3 sensors-20-06113-f003:**
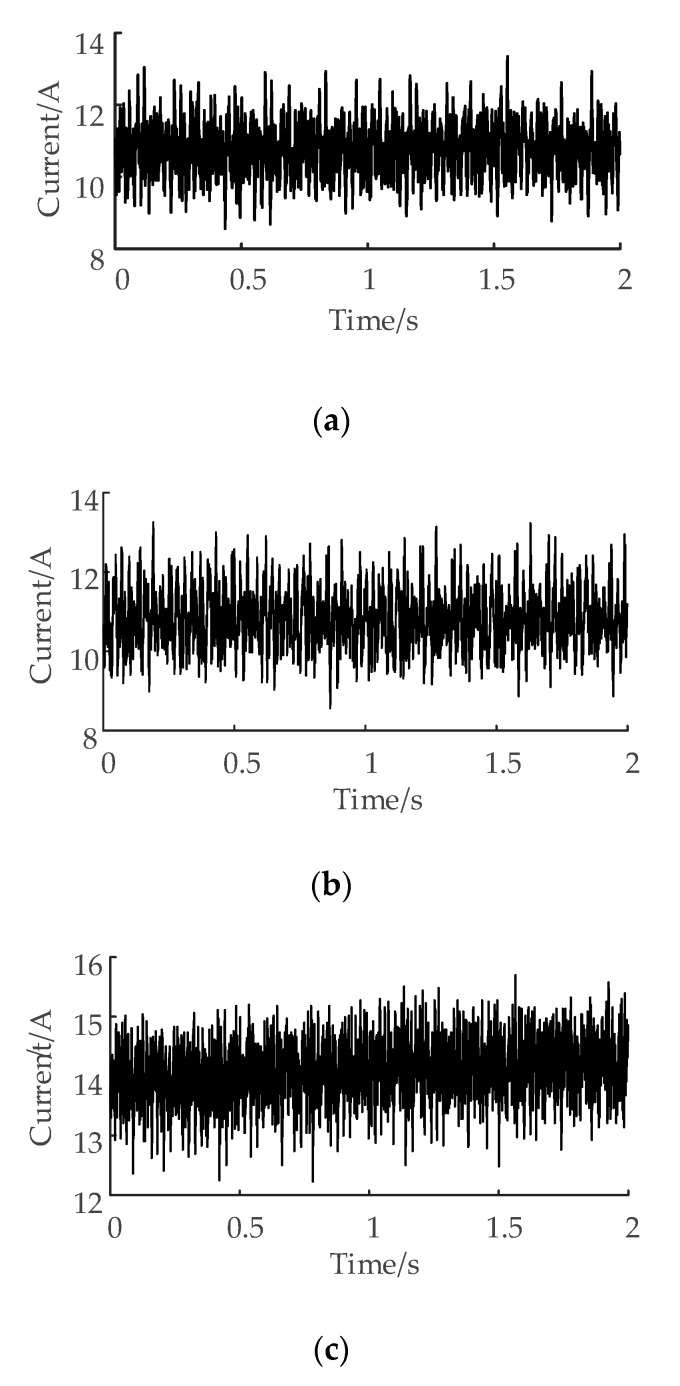
(**a**) Time–domain waveform of the current signal at the initial tool wear stage, (**b**) time–domain waveform of the current signal at the normal tool wear stage, and (**c**) time–domain waveform of the current signal at the sharp tool wear stage.

**Figure 4 sensors-20-06113-f004:**
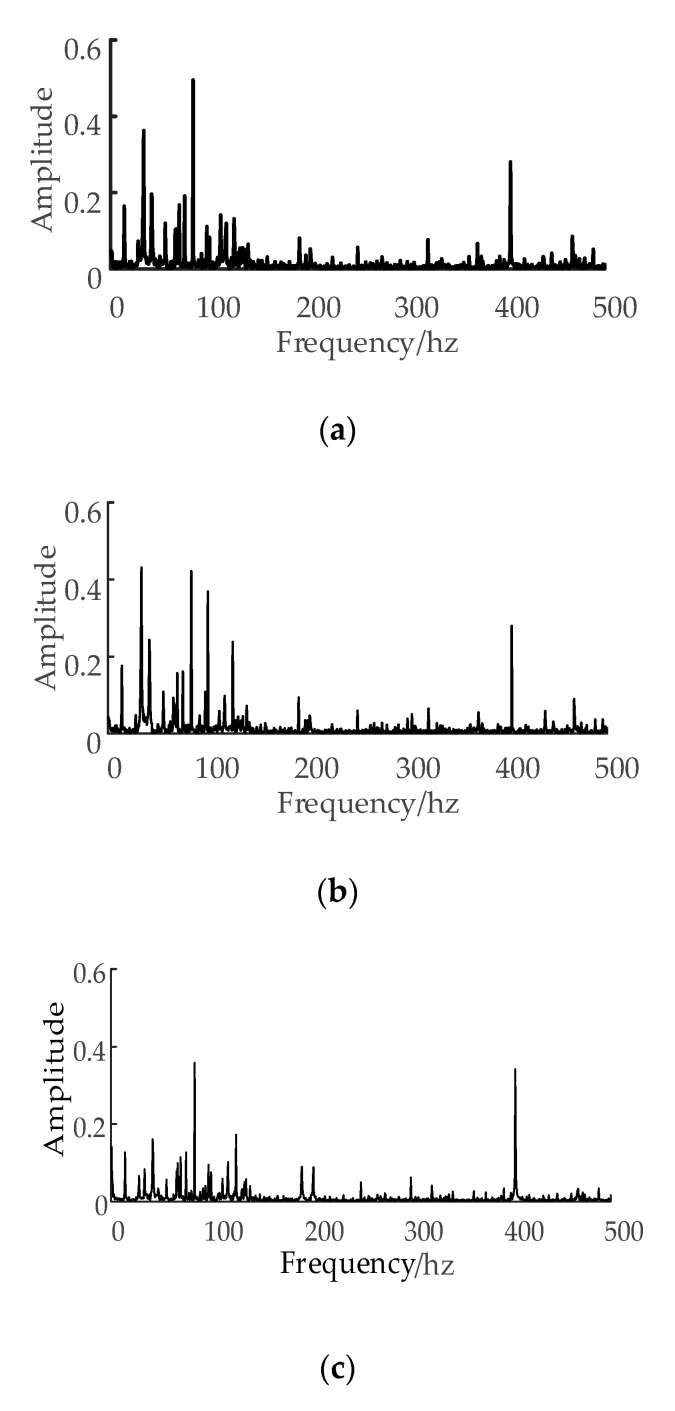
(**a**) Amplitude-frequency chart of the current signal at the initial tool wear stage, (**b**) the normal tool wear stage, and (**c**) the sharp tool wear stage.

**Figure 5 sensors-20-06113-f005:**
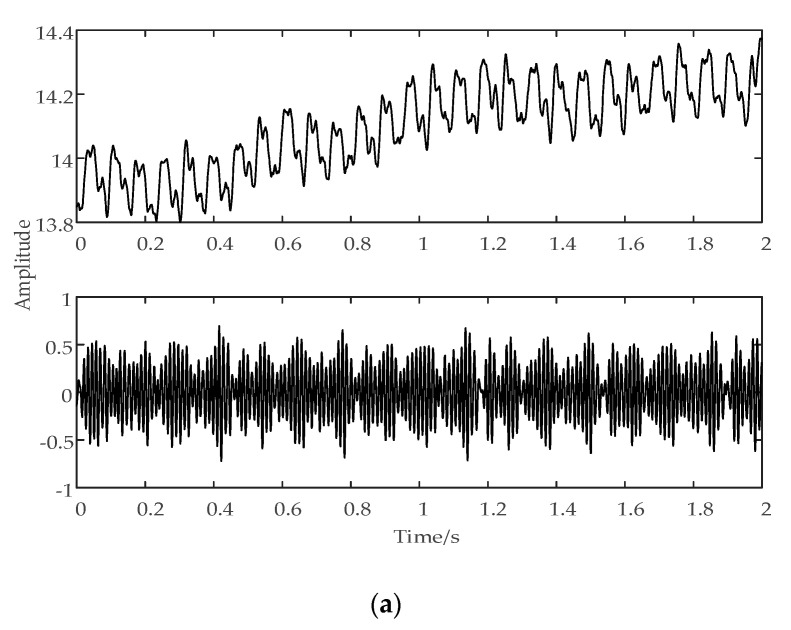
The variational mode decomposition (VMD) result of the current signal under different K values: (**a**) K = 2, (**b**) K = 3, (**c**) K = 4, (**d**) K = 5, (**e**) K = 6, and (**f**) K = 7.

**Figure 6 sensors-20-06113-f006:**
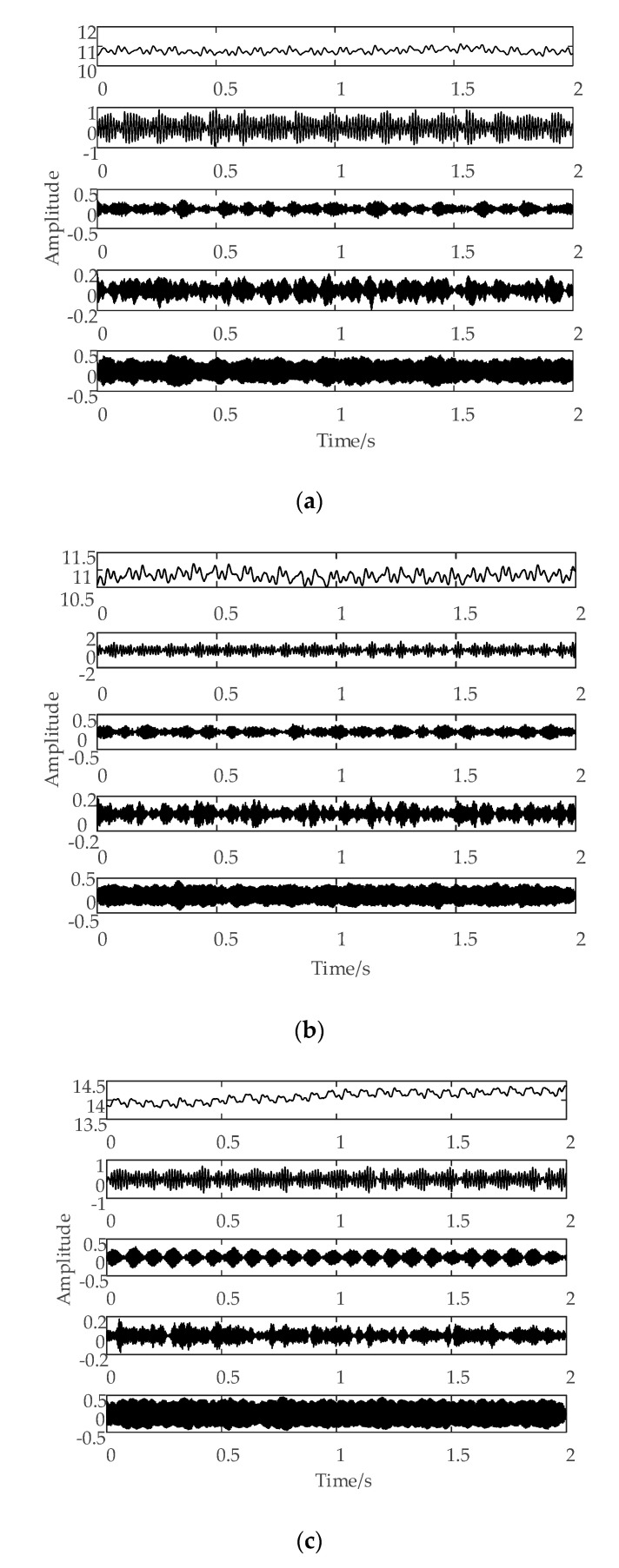
(**a**) The VMD result of the current signal at the initial tool wear stage, (**b**) the normal tool wear stage, and (**c**) the sharp tool wear stage.

**Figure 7 sensors-20-06113-f007:**
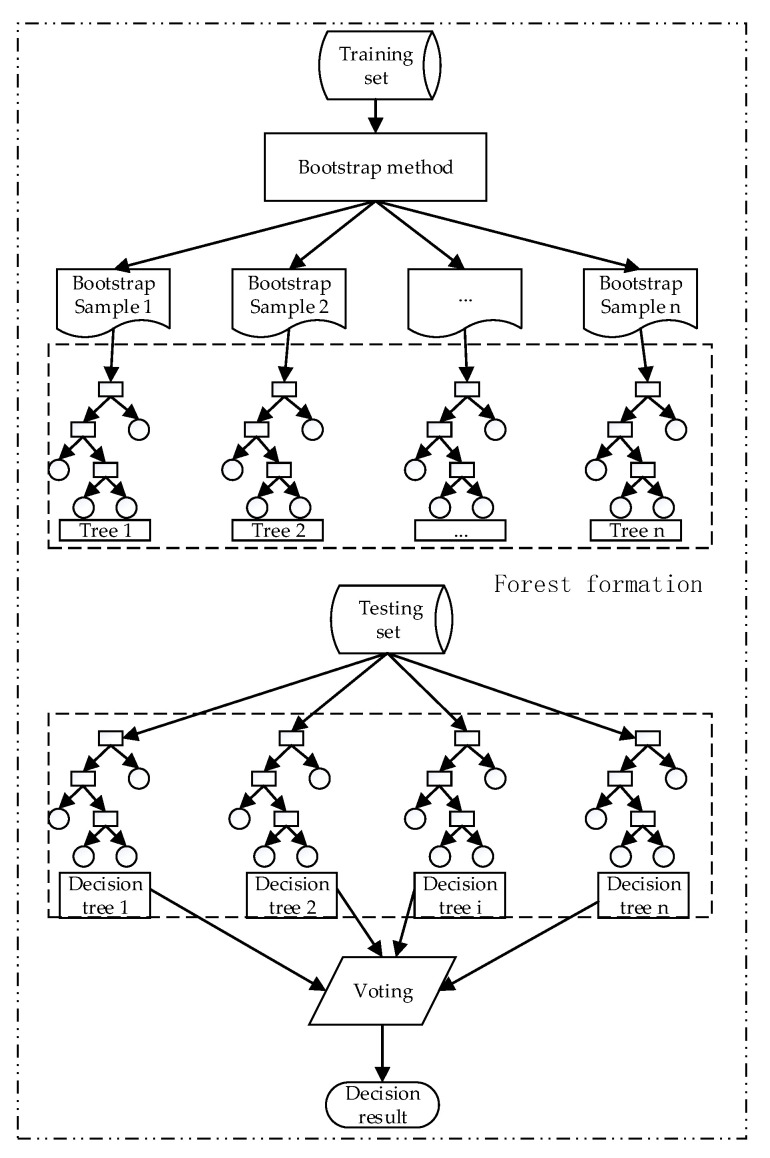
Schematic diagram of the random forest (RF) algorithm.

**Figure 8 sensors-20-06113-f008:**
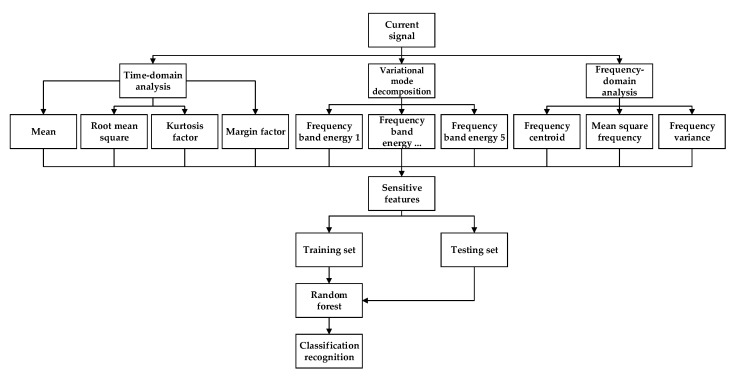
Tool wear condition recognition roadmap based on current signal analysis.

**Figure 9 sensors-20-06113-f009:**
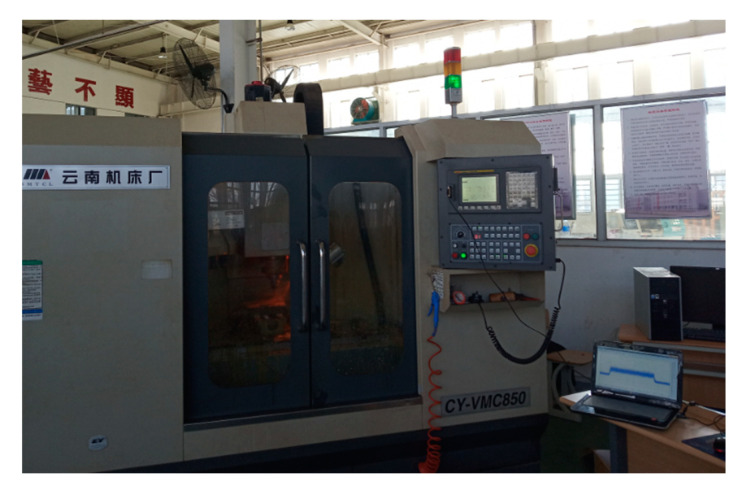
Experimental setup.

**Figure 10 sensors-20-06113-f010:**
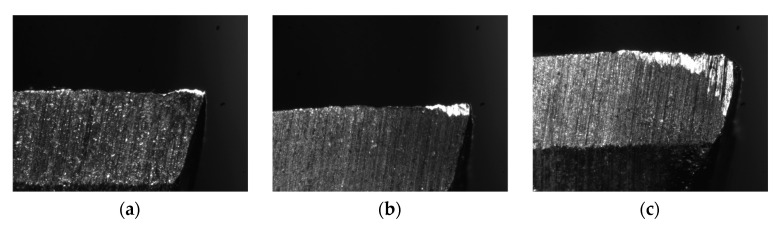
(**a**) The tool image of the initial wear stage, (**b**) the normal wear stage, and (**c**) the sharp wear stage.

**Figure 11 sensors-20-06113-f011:**
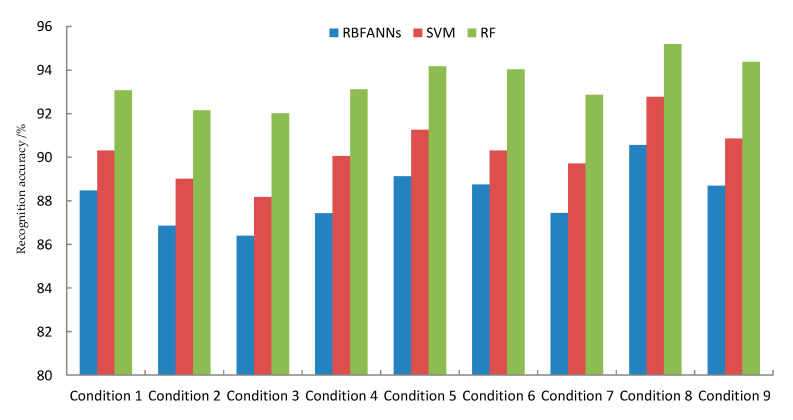
The predicted recognition accuracy under different conditions with different classifiers. RBFANNs: radial basis function artificial neural networks, SVM: support vector machines, and RF: random forest.

**Figure 12 sensors-20-06113-f012:**
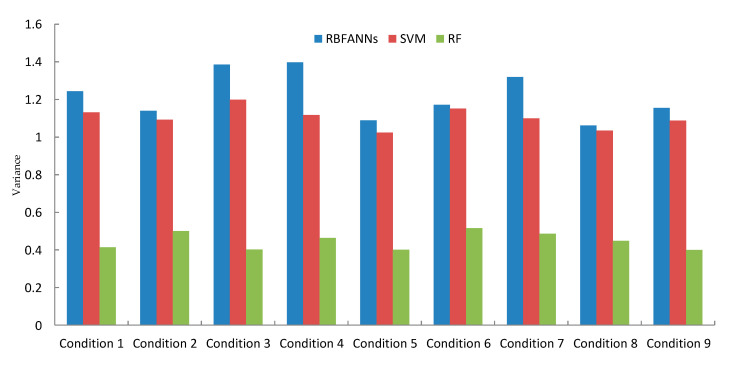
The predicted result variance under different conditions with different classifiers.

**Table 1 sensors-20-06113-t001:** The central frequency corresponding to different K values.

Mode Number (K)	Central Frequency/Hz						
2	0	194	-	-	-	-	-
3	0	82	405	-	-	-	-
4	0	82	195	405	-	-	-
5	0	82	195	301	405	-	-
6	0	82	195	252	302	405	-
7	0	81	123	196	301	404	405

**Table 2 sensors-20-06113-t002:** List of extracted features.

Feature Extraction Method	Signal Features	Expression
Time–domain analysis	Mean ($x_{i}$)	$\mu =E\left(\left|x_{i}\right|\right)$
	Root mean square (RMS)	$x_{\mathrm{RMS}}={\left\{E\left(x_{i}^{2}\right)\right\}}^{1/2}$
	Kurtosis factor (Kur)	$x_{Kur}=E\left\{{\left[\left(\left|x_{i}\right|\right)-\mu /\sigma \right]}^{4}\right\}$ $\sigma ={\left\{E\left[{\left(\left|x_{i}\right|-\mu \right)}^{2}\right]\right\}}^{1/2}$
	Margin factor (Mar)	$x_{\mathrm{Mar}}=x_{\mathrm{mas}}/{\left(\frac{1}{n}\overset{n}{\underset{i=1}{\sum }}{\left|x_{i}\right|}^{1/2}\right)}^{2}$
Frequency–domain analysis	Frequency centroid (FC)	$x_{FC}=\frac{\sum_{i=1}^{N}f_{i}\cdot p\left(f_{i}\right)}{\sum_{i=1}^{N}p\left(f_{i}\right)}$
	Mean square frequency (MSF)	$x_{MSF}=\frac{\sum_{i=1}^{N}f_{i}{}^{2}\cdot p\left(f_{i}\right)}{\sum_{i=1}^{N}p\left(f_{i}\right)}$
	Frequency variance (FV)	$x_{FV}=\frac{\sum_{i=1}^{N}{\left(f_{i}-x_{Fc}\right)}^{2}\cdot p\left(f_{i}\right)}{\sum_{i=1}^{N}p\left(f_{i}\right)}$
Variational mode decomposition	Frequency band energy	$E_{i}=\overset{N}{\underset{k=1}{\sum }}{\left|x_{ik}\right|}^{2}\left(i=1,2,3,4,5\right)$

**Table 3 sensors-20-06113-t003:** Operating conditions.

Test Group	Spindle Speed (r/min)	Cutting Depth (mm)	Feed Rate (mm/min)	Cutting Condition
Condition 1	1000	0.5	500	Dry milling
Condition 2	1000	1	650	Dry milling
Condition 3	1000	1.5	800	Dry milling
Condition 4	1500	0.5	650	Dry milling
Condition 5	1500	1	800	Dry milling
Condition 6	1500	1.5	500	Dry milling
Condition 7	2000	0.5	800	Dry milling
Condition 8	2000	1	500	Dry milling
Condition 9	2000	1.5	650	Dry milling

**Table 4 sensors-20-06113-t004:** Division of the tool wear stages. VB: maximum width of the wear land.

Wear Stage	VB/mm
Initial wear stage	0 ≤ VB < 0.2
Normal wear stage	0.2 ≤ VB < 0.5
Sharp wear stage	VB ≥ 0.5

**Table 5 sensors-20-06113-t005:** The recognition accuracy of the tool wear condition based on time–domain analysis, frequency–domain analysis, and wavelet packet decomposition (WPD).

Number	Accuracy			Average Recognition Accuracy	Overall Accuracy	Variance
	Initial	Normal	Sharp			
1	85.19%	90.12%	84.72%	86.68%	87.45%	1.1363
2	82.61%	87.78%	94.03%	88.14%		
3	93.59%	69.70%	92.75%	85.35%		
4	77.42%	95.60%	86.21%	86.41%		
5	83.87%	87.06%	95.31%	88.75%		
6	83.33%	89.47%	93.24%	88.68%		
7	89.66%	78.82%	96.97%	88.48%		
8	77.42%	90.48%	95.35%	87.75%		
9	85.19%	85.53%	89.61%	86.78%		
10	83.33%	88.37%	90.63%	87.44%		

**Table 6 sensors-20-06113-t006:** The recognition accuracy of the tool wear condition based on the time–domain analysis, frequency–domain analysis, and variational mode decomposition (VMD).

Number	Accuracy			Average Recognition Accuracy	Overall Accuracy	Variance
	Initial	Normal	Sharp			
1	94.74%	98.61%	97.14%	96.83%	95.19%	0.4483
2	97.06%	94.52%	94.52%	95.37%		
3	96.67%	93.83%	95.65%	95.38%		
4	100%	91.67%	94.81%	95.49%		
5	96.97%	97.33%	91.67%	95.32%		
6	100%	90.70%	94.03%	94.91%		
7	96.43%	96.00%	92.21%	94.88%		
8	93.10%	90.80%	98.44%	94.11%		
9	92.59%	97.65%	94.11%	94.78%		
10	91.43%	96.00%	97.14%	94.86%		

**Table 7 sensors-20-06113-t007:** The recognition accuracy under different operating conditions using WPD.

Test Group	Recognition Accuracy	Variance
Condition 1	85.19%	1.1771
Condition 2	85.13%	1.1205
Condition 3	86.04%	1.4022
Condition 4	83.46%	1.3674
Condition 5	85.38%	1.2998
Condition 6	84.07%	1.1074
Condition 7	86.28%	1.3063
Condition 8	87.45%	1.1363
Condition 9	85.36%	1.4042

**Table 8 sensors-20-06113-t008:** The recognition accuracy under different operating conditions using VMD.

Test Group	Recognition Accuracy	Variance
Condition 1	93.07%	0.4138
Condition 2	92.15%	0.5001
Condition 3	92.02%	0.4019
Condition 4	93.12%	0.4636
Condition 5	94.18%	0.4001
Condition 6	94.04%	0.5152
Condition 7	92.87%	0.4868
Condition 8	95.19%	0.4483
Condition 9	94.38%	0.3998

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
