# Peer review of "Tool Wear Condition Monitoring by Combining Variational Mode Decomposition and Ensemble Learning"

_sensors, 2020, doi:10.3390/s20216113_

Round 1
Reviewer 1 Report
Responses to the authors' comments:
"3. Response: Although the random forest algorithm has been applied in other fields and the results of its application is satisfactory, it is rarely used in the classification of tool wear condition. In this paper, the random forest algorithm is applied to the classification of tool wear condition, and the experimental results show that the random forest algorithm has higher recognition accuracy, smaller variance and better robustness compared with a single classifier in the recognition of tool wear condition."
Although the authors pointed out that the random forest is rarely used in the classification of tool wear condition, it is not difficult to find such a study.
For examples,
[1] "Wu, D., Jennings, C., Terpenny, J., Gao, R. X., and Kumara, S. (April 18, 2017). "A Comparative Study on Machine Learning Algorithms for Smart Manufacturing: Tool Wear Prediction Using Random Forests." ASME. J. Manuf. Sci. Eng. July 2017; 139(7): 071018. https://doi.org/10.1115/1.4036350"
[2] "Pimenov, D.Y., Bustillo, A. & Mikolajczyk, T. Artificial intelligence for automatic prediction of required surface roughness by monitoring wear on face mill teeth. J Intell Manuf 29, 1045–1061 (2018). https://doi.org/10.1007/s10845-017-1381-8"
Therefore, it is still unclear to say that the major work (3) written in this paper (lines 479-483) has an academic value.
Author Response
Dear reviewer:
On behalf of my co-authors, we thank you very much for giving us an opportunity to revise our manuscript.
Point: Although the authors pointed out that the random forest is rarely used in the classification of tool wear condition, it is not difficult to find such a study. Therefore, it is still unclear to say that the major work (3) written in this paper (lines 479-483) has an academic value.
Response: According to your precious advice, we believe that the previous summary of major work (3) is not appropriate. We should talk about what we do. So we have made a new summary in the revised manuscript (lines 481-485) after much discussion.
We have tried our best to revise our manuscript according to the comment. Attached please find the revised version, which we would like to submit for your kind consideration.
Once again, thank you very much for your comments and suggestions.
Thank you and best regards.
Yours sincerely,
Jun Yuan
Reviewer 2 Report
All previous comments and suggestions were addressed in a satisfactory way. There are still a few issues in the written English that should be improved and affect readability.
Author Response
Dear reviewer:
On behalf of my co-authors, we thank you very much for giving us an opportunity to revise our manuscript.
Point: All previous comments and suggestions were addressed in a satisfactory way. There are still a few issues in the written English that should be improved and affect readability.
Response: According to your precious advice, we revised our manuscript carefully with the help of MDPI English Editing Office. We have tried our best to revise our manuscript according to the comment. Attached please find the revised version, which we would like to submit for your kind consideration.
Once again, thank you very much for your comments and suggestions.
Thank you and best regards.
Yours sincerely,
Jun Yuan
Reviewer 3 Report
The authors have addressed my earlier concerns, and as such, I recommend the paper to be accepted for publication.
Author Response
Dear reviewer:
Thanks very much for your kind work and consideration on our paper. On behalf of my co-authors, we would like to express our great appreciation to you. Once again, thank you very much for your comments and suggestions.
Thank you and best regards.
Yours sincerely,
Jun Yuan
This manuscript is a resubmission of an earlier submission. The following is a list of the peer review reports and author responses from that submission.
Round 1
Reviewer 1 Report
This paper is about tool-wear classification using the motor current signal of a machining center's spindle. However, the reviewer could not find academic improvements in this paper.
As the major work of this study, the authors mentioned three items ((1)-(3)) summarised in the conclusions. But, regarding the item (1), many conventional studies have already confirmed the correlation between the motor current and tool-wear. The authors should cite them properly and state the difference clearly. Second, although this study considered that a variational-mode-decomposition approach (VMD) is better than a wavelet-packet-decomposition (WPD), which was written as the item (2), the cutting experiments were executed under only the specific condition. So, this point could not be validated yet under general conditions. Lastly, the superiority of the RF (random forest) algorithm, which was listed as the item (3), was also shown conventionally, as the authors described it on page 13 as well. Therefore, it is not easy to evaluate these three items as the originality of this paper, and the reviewer believes that this paper does not reach the publication quality of the high-impact journal.
Author Response
1. Response to comment: (Many conventional studies have already confirmed the correlation between the motor current and tool-wear. The authors should cite them properly and state the difference clearly.)
Response: According to your precious advice, we have deleted Part 2 (The Correlation Mechanism between Spindle Motor Current and Tool Wear Condition) of the original text, and cited it in the introduction part. In addition, we have made a comparative analysis. (line 81-101).
2. Response to comment: (Although this study considered that a variational-mode-decomposition approach (VMD) is better than a wavelet-packet-decomposition (WPD), which was written as the item (2), the cutting experiments were executed under only the specific condition. So, this point could not be validated yet under general conditions.)
Response: According to your precious advice, we have added cutting experiments under multi-conditions. Part 3.2 (Results and Discussions) of the newly submitted paper show that the VMD has an excellent performance in processing the non-stationary current signal, and the frequency band-energy features obtained by the VMD of current signal are more sensitive to tool wear than that obtained by the frequently-used WPD of current signal (line 380-381, line 438-445).
3. Response to comment: ( The superiority of the RF (random forest) algorithm, which was listed as the item (3), was also shown conventionally, as the authors described it on page 13 as well.)
Response: Although the random forest algorithm has been applied in other fields and the results of its application is satisfactory, it is rarely used in the classification of tool wear condition. In this paper, the random forest algorithm is applied to the classification of tool wear condition, and the experimental results show that the random forest algorithm has higher recognition accuracy, smaller variance and better robustness compared with a single classifier in the recognition of tool wear condition.
Special thanks to you for your good comments.
Reviewer 2 Report
Line 2 – Title- It is not common the use of the word “recognition” – “Monitoring” is universally accepted and should be used – why recognition? please explain.
Line 18 – It reads “ FANUC SEVRO GUIDE” – I think it should be “FANUC SERVO GUIDE”; This is repeated throughout the manuscript and should be verified! There are 8 occurrences of “SEVRO”!
Line 37 – It is said that “tool failure 36 leads to at least 20% of unscheduled downtime in modern manufacturing systems [1].” – the reference used is from 1997, it is outdated and does not support the argument.
Line 40 – Again the references are too old!
Line 77 – A deeper review of past literature on spindle current condition monitoring should be given since this has already been done by many researchers.
Line 83 – In this formula there is a reference to dw/dt – why? … In the context of machining!? … I see the answer in Line 95 but this is obvious!
Line 86 – The reference to “frequency” is not correct, please rectify.
Line 88-91 – There is a reference to a formula to calculate the RMS current, first why and what for? Is this formula even correct?!
Line 119 – A reference is made to “upper computer” but there is no explanation for this!
Line 115-122 – There is an explanation on how to acquire the current from the software/hardware but little is provided – since it is now in the research focus should be eliminated. 127-133 again little is added … just say that it is acquired directly from the machine without making changes to the machine or interfering with the cutting process.
Line 170 – graphs in figure would benefit from a normalized scale – same scale to be comparable.
Line 372 – It is stated that according to relevant standards --? .. what standards? … a flank wear level above 0.3 mm is considered worn! .. and there is an ISO standard for that! Please clarify.
Line 382 – The set of features extracted is repeated and described, as in Line 160.
The methods used to extract information, and corresponding review, (features) are scattered along the paper – a section should be built where all literature review is conducted. Also, some experimental parts/data are presented in between some of the review.
English writing was not thoroughly checked and should be extensively revised – there are some sentences that should be rewritten: Line 12 first sentence of the abstract; Line 21 “the sensitive features is substituted 21 into the ensemble learning…” … the all abstract needs an in-depth English review; Line 33 “As an essential component of advanced manufacturing system, tool plays an important role in machining.” This sentence lacks readability; Line 39 – The paragraph starting here has many writing mistakes and very poor writing! …. A lot to revise in between … Line 99 “will be got” what is it meant?! …All the text should be revised by native English professionals.
Author Response
- Response to comment:( Line 2 – Title- It is not common the use of the word “recognition” – “Monitoring” is universally accepted and should be used – why recognition? please explain.)
Response: According to your precious advice, we believe that monitoring should be used instead of recognition after much discussion.
- Response to comment:( It reads “ FANUC SEVRO GUIDE” – I think it should be “FANUC SERVO GUIDE”; This is repeated throughout the manuscript and should be verified! There are 8 occurrences of “SEVRO”!)
Response: I am sorry for my negligence. It is “FANUC SERVO GUIDE”, and we have corrected it.
- Response to comment:( It is said that “tool failure 36 leads to at least 20% of unscheduled downtime in modern manufacturing systems [1].” – the reference used is from 1997, it is outdated and does not support the argument.)
Response: We have cited an apt reference in the introduction part (line 491-492).
- Response to comment:( Line 40 – Again the references are too old!)
Response: We have cited an apt reference in the introduction part (line 493-495).
- Response to comment:( A deeper review of past literature on spindle current condition monitoring should be given since this has already been done by many researchers.)
Response: According to your precious advice, we have deleted Part 2 (line 77-112) of the original text, and cited it in the introduction part. In addition, we have made a comparative analysis. Methods analysis and experimental results show that our proposed method is more convenient and effective in practical application (line 80-101).
- Response to comment:( A reference is made to “upper computer” but there is no explanation for this!)
Response: The upper computer here refers to the computer used to receive the spindle current signal (line 169-170).
.Response to comment:( Line 115-122 – There is an explanation on how to acquire the current from the software/hardware but little is provided – since it is now in the research focus should be eliminated. 127-133 again little is added … just say that it is acquired directly from the machine without making changes to the machine or interfering with the cutting process.)
Response: I am sorry for the above problems. We add a detailed explanation to the second part (2.1.The Method of Spindle Motor Current Signal-acquisition) of the paper (line 169-178).
- Response to comment:( graphs in figure 4 would benefit from a normalized scale – same scale to be comparable.)
Response: According to your precious advice, we have made a normalized scale in Figure 4 (line 233).
- Response to comment:( It is stated that according to relevant standards --?. what standards? … a flank wear level above 0.3 mm is considered worn! .. and there is an ISO standard for that! Please clarify.)
Response: Since the material of workpiece and cutter used in the experiment are low carbon steel and high-speed steel respectively, and the processing condition is rough machining, the tool flank's maximum width of the wear land above 0.5 mm is considered worn. In addition, the purpose of this section is to evaluate the performance of the classifier. So the small difference of wear stage division does not affect the evaluation. But you are right. I should not use the phrase “relevant standards” here. I should explain according to above view (line 394-396).
- Response to comment:( The set of features extracted is repeated and described, as in Line 160.)
Response: We have corrected it in part 2.2.4 (line 335).
- Response to comment:( The methods used to extract information, and corresponding review, (features) are scattered along the paper – a section should be built where all literature review is conducted. Also, some experimental parts/data are presented in between some of the review.)
Response: According to your precious advice, we have redesigned the above contents in the resubmitted paper.
- Response to comment:( English writing was not thoroughly checked and should be extensively revised)
Response: We have tried our best to revise our manuscript according to the comments. We would like to submit for your kind consideration.
Special thanks to you for your good comments.
Reviewer 3 Report
In this paper, the authors proposed using current signal for tool condition monitoring, as this does not interfere with the machining process itself, which happens with for e.g. force measurement. The current data is then analysed using variational mode decomposition (VMD) to extract certain features, which are in turn fed through an ensemble learning (EL) classifier for tool condition classification. Experimental results show that the proposed method outperforms wavelet packet decomposition (WPD) as well as RBF networks and SVM.
The paper is easily readable. However, there are some improvements required before the paper can be accepted / published:
- The current measurement works easily only on the FANUC machine with SEVRO package. There will be quite substantial work needed for other machines. Can the authors please comment on this?
- There are many papers which proposed using current for tool condition monitoring. How does the result of this paper compare with the others?
- The algorithm in lines 218 to 222 should be put in a "box".
- Please explain figure 5 in more details. Some guesswork is needed right now to interpret the graphs.
- In lines 275-279, please explain what "mode mixing" means and how we can see them from figure 5.
- Explain "VB" in table 3.
- In the paragraph starting from line 382, there is a long explanation about what means, RMS, kurtosis etc. mean. I think these should be explained earlier in section 3.2.
- In the experiment, the proposed method is compared with WPD. Why did the authors choose WPD? This was not mentioned in the literature review.
- Some comparison with other literature using only current, or current + other measurements e.g. acoustic emission should be provided.
Author Response
1. Response to comment:( The current measurement works easily only on the FANUC machine with SEVRO package. There will be quite substantial work needed for other machines. Can the authors please comment on this?)
Response: The internal system of different types of machine tools is basically equipped with embedded Ethernet board card. Users can easily connect it with the upper computer through the network cable to obtain the current data. Especially for some open numerical control machine tools, the user can easily read the current data of the built-in current sensor through the secondary development protocol. For the old machine tools which are not open to the users, users can obtain the current data of the spindle through the external current sensor which does not affect the actual processing.
2. Response to comment:( There are many papers which proposed using current for tool condition monitoring. How does the result of this paper compare with the others?)
Response: According to your precious advice, we add a comparative analysis with them in the introduction part (line 80-101). Methods analysis and experimental results show that our proposed method is more convenient and effective in practical application.
3. Response to comment:( The algorithm in lines 218 to 222 should be put in a "box".)
Response: We have put them in a “box”(line 276-281).
4. Response to comment:( Please explain figure 5 in more details. Some guesswork is needed right now to interpret the graphs.)
Response: I am sorry for the above problem; we add a detailed explanation (line 311-319) in the resubmitted paper.
5. Response to comment:( In lines 275-279, please explain what "mode mixing" means and how we can see them from figure 5.)
Response: According to your precious advice, we add an explanation (line 286-288) for above problems.
6. Response to comment:( Explain "VB" in table 3.)
Response: I am sorry for my negligence. We have added an explanation (line 395-396) for” VB”.
7. Response to comment:( In the paragraph starting from line 382, there is a long explanation about what means, RMS, kurtosis etc. mean. I think these should be explained earlier in section 3.2.)
Response: According to your precious advice, we have redesigned the above contents in the resubmitted paper (line 336-354).
8. Response to comment:( In the experiment, the proposed method is compared with WPD. Why did the authors choose WPD? This was not mentioned in the literature review.)
Response: According to your precious advice, we make an analysis why many authors choose WPD in the introduction part (line 102-104).
9. Response to comment:( Some comparison with other literature using only current, or current + other measurements e.g. acoustic emission should be provided. )
Response: According to your precious advice, the performance of the proposed method for tool wear condition monitoring is analyzed and compared with other literature using only current, or current + other measurements e.g. acoustic emission (line 78-101).
Special thanks to you for your good comments.